# Management Strategies for Common Animal Bites in Pediatrics: A Narrative Review on the Latest Progress

**DOI:** 10.3390/microorganisms12050924

**Published:** 2024-05-01

**Authors:** Dragos Septelici, Giulia Carbone, Alessandro Cipri, Susanna Esposito

**Affiliations:** Pediatric Clinic, Department of Medicine and Surgery, University of Parma, 43126 Parma, Italy; dragos.septelici@unipr.it (D.S.); giulia.carbone@unipr.it (G.C.); alessandro.cipri@unipr.it (A.C.)

**Keywords:** animal bite, antimicrobial therapy, pediatric infectious disease, rabies, tetanus

## Abstract

Animal bites are a common reason for children to visit primary care and emergency departments. Dog bites are the most prevalent, followed by cat bites at 20–30%. Other animals such as bats, monkeys, snakes, and rats collectively contribute less than 1% of cases. Hospitalization is necessary in only 4% of animal bite incidents. The main aim of this narrative review is to summarize the main protocols currently followed in pediatrics in cases involving the most common bites from different animal species. Analysis of the literature showed that the management of common animal bites in children presents a multifaceted challenge requiring a comprehensive understanding of the epidemiology, clinical presentation, and treatment modalities associated with each specific species. Effective wound management is paramount in reducing the risk of infection and promoting optimal healing outcomes. Additionally, tetanus vaccination status should be assessed and updated as necessary, and prophylactic antibiotics may be indicated in certain cases to prevent secondary infections. Furthermore, the role of rabies prophylaxis cannot be overstated, particularly in regions where rabies is endemic or following bites from high-risk animals. In addition to medical management, psychosocial support for both the child and their caregivers is integral to the overall care continuum. Future studies exploring the efficacy of novel treatment modalities, such as topical antimicrobial agents or advanced wound dressings, may offer new insights into optimizing wound healing and reducing the risk of complications.

## 1. Introduction

Animal bites represent a frequent cause of primary care and emergency department visits by children [1]. Although epidemiological data showed that about 0.2% of all emergency room admissions are related to this problem [2], this information is vastly underestimated because bites are often judged to be minor, and, very frequently, children are not brought to medical attention. However, only 4% of animal bites require hospitalization, and 99% of injuries are considered to be low risk [3]. In most cases, these are dog bites (60–80%), followed by cat bites (20–30%); other animals (e.g., bats, monkeys, snakes, and rats) account for less than 1% of cases [4]. In the United States, about 800,000 dog bites are treated each year, for a total cost of about $30 million per year [5]. Children are more likely to be the victim of animal bites than are adults. As for the site, bite wounds on the face account for only 10% of the cases, and most of these (about 70%) occur in children under the age of 10 years; in older children, instead, the most frequently affected site is the hand [1].

Despite the commonalities of animal bites, the approaches to managing these injuries can vary widely. Factors such as the severity of the bite, the animal species involved, and the child’s health status all influence the course of treatment [1,2,3]. Moreover, appropriate management strategies are crucial not only for addressing immediate medical concerns but also for preventing long-term complications such as infection, scarring, and psychological trauma. Furthermore, it is important to remember that the majority of animal bites in children are preventable, which highlights the critical roles of education and proactive measures in safeguarding their well-being [1]. Preventive efforts encompass various aspects, including education on animal behavior, responsible pet ownership, and avoidance tactics when encountering unfamiliar animals. Understanding the importance of prevention strategies is essential for healthcare providers, policymakers, educators, and the community at large.

In this narrative review our goal was to describe the state of the art regarding the main protocols currently followed in cases of the most common bites from animal species in children. The MEDLINE/PubMed database was searched from 1993 to 29 February 2024 to collect the relevant literature. The search included randomized placebo-controlled trials, controlled clinical trials, double-blind, randomized controlled studies, and systematic reviews and meta-analyses. Abstracts were excluded. The following combinations of keywords were used: “animal” or “cat” or “dog” or “monkey” or “snake” or “bat” or “rat” and “children” or “pediatric” or “paediatric” and “bite”. We selected only articles written in English, and we performed a manual search based on the references of suitable articles.

## 2. General Management of Animal-Bite Wounds

Animal-bite wounds are almost always contaminated; therefore, adequate treatment is often required to prevent any secondary infections [6]. They can be classified as avulsions, lacerations, or punctures. Infectious risk can vary depending on the animal involved, as seen by an up to 50% risk for cat bite and a risk from 5% to 20% for dog bite [4,7]. First, it is important to adequately clean the wound with clean or saline water and perform high pressure irrigations with saline or aseptic solution with at least a 20-mL syringe or a 20-gauge catheter [4,6]. It should be remembered to avoid using hydrogen peroxide or alcohol, as they can be too harsh and may delay healing. Very contaminated wounds require more irrigation, while large and dirty ones require irrigation in the operating room [8]. In case of wound infection, it is important to perform a specific bacterial detection by culture examination [1]. Subsequently, it is necessary to remove the devitalized tissue and any foreign material, and check for possible involvement of internal tissues, such as bones or tendons [4]; in cases in which those internal tissues are involved, diagnostic imaging needs to be performed [1]. Next, the wound should be covered with a sterile, non-stick dressing pad to protect it from further contamination and promote a moist healing environment [1]. The dressing should be closed with medical tape or a bandage, ensuring that it is not too tight to allow proper circulation [1]. Finally, it is essential to monitor the wound for signs of infection, such as increased redness, swelling, warmth, or drainage. If any signs of infection occur or if the wound does not seem to be healing properly, it is crucial to seek medical attention promptly.

The decision as to when to close the wounds is still controversial. Current guidelines from The Infectious Diseases Society of America (IDSA) state that initial wound closure is not recommended, except for facial wounds; in this case it is important to carry out adequate cleaning and debridement and ensure sufficient antibiotic coverage [9]. On the other hand, bites located in other sites are normally left open due to the potential infectious risk [6]. Wound closure certainly improves the subsequent aesthetic appearance, but it could increase the risk of infection [10]. However, data in the literature are still conflicting. Some studies have found no difference between patients who had their wound closed versus those with wounds left open. Jaindl et al. observed in their study that the risk of secondary infection was made nearly equal by adequately treating wounds prior to closure [11]. Moreover, a study by Chen et al. showed that the risk of infection, comparing the two methods, has overlapping results (7.6% vs. 7.8%) [10]. However, Paschos et al. confirmed an increased risk of infection by primary closing compared to healing by secondary intention [12]. As a general criterion, suturing the wound is still contraindicated in puncture and deep wounds, such as cat bites [4]. Deeper lesions require surgical consultation for subcutaneous suturing, while patients with avulsion of facial skin need a consultation with the plastic surgeon, who can evaluate possible use of grafts or skin flap repair [13]. 

Antibiotic prophylaxis is recommended for all wounds at high risk of infection, like moderate to severe injuries on the hands, feet, face, and genitals; bites with signs of infection, puncture, and deep wounds; cases where the victim is affected by diabetes mellitus or immunosuppression; injuries that may have penetrated the periosteum or joint capsule; and cat bites [9]. On the contrary, antibiotics should not be administered if there are no clinical signs of infection in the first 24 h after the animal bite [7]. Since bacterial cultures are not readily available, there is a need to practice empirical therapy: first-line treatment for children is amoxicillin-clavulanic acid at the dosage of 50 mg/kg twice daily, which protects against both aerobic and anaerobic bacteria [9]. The duration of the antibiotic therapy, in the absence of clear signs of infection, is about 3 to 5 days, but children need to be evaluated over the next two days to see if any sign of infection has appeared [1]. If practiced, the antibacterial therapy should be modified based on the results of the culture isolation. In children with penicillin allergy, an alternative therapeutic option is clindamycin plus trimethoprim-sulfamethoxazole or an extended-spectrum oral cephalosporin in conjunction with clindamycin to provide additional protection against anaerobes [1,14]. In cases in which there is a need of intravenous therapy, ampicillin-sulbactam is recommended [7]. Due to the spread of methicillin-resistant *Staphylococcus aureus* (MRSA), a coverage with trimethoprim-sulfamethoxazole, doxycycline, or clindamycin should be considered in children with severely infected wounds [1]. 

Another important aspect of the general management of animal bites is to consider the patient’s immunization status against tetanus, by checking the number of vaccine doses received and the time since the last dose, because in children who have received less than three doses, both vaccination and tetanus-specific immunoglobulins are recommended, while in cases of the administration of three or more doses when it has been less than five years since the last dose, no treatment should be given [6,15]. Table 1 describes schematic antitetanic management in Italy according to the Italian Health Ministry. 

Only vaccine administration is required if more than 5 years have passed. Both anti-tetanus immunoglobulins and the vaccine are administered if at least 10 years have passed. In cleaner and smaller wounds, vaccination alone is sufficient. Anti-tetanus immunoglobulins should be administered intramuscularly at a dosage of 250 IU, regardless of age or weight [6,15,16].

Another important aspect is represented by anti-rabies prophylaxis, which depends on the animal species involved and whether or not the animal comes from a rabies-free geographic area (Table 2). Post-exposure prophylaxis (PEP) consists of vaccine administration, combined, or not, with rabies-specific immunoglobulins (RIGs). Vaccine administration is by the intramuscular route. RIGs are administered intradermally at a dosage of 20 IU/Kg, regardless of age, within 7 days from the first vaccine dose. According to the 2018 WHO guidelines [17], if patients may be classified as potentially risky or high-risk exposure, the vaccine is always recommended. In these cases, the first dose of vaccine is given on day 0, additional doses on days 3 and 7, and a fourth dose between days 14 and 28; RIGs are recommended in this case. If the patient was previously immunized, only two doses are given, at days 0 and 3 post-exposure, and no RIGs are administered [17]. In addition, the sites of biting should be taken in consideration during the handling of cases, i.e., bites near the brain should be treated immediately. 

Controlling biting animals is essential for preventing incidents of animal bites and minimizing risks to public health and safety. Various measures are employed to manage and control populations of biting animals, including both domestic pets and wildlife [7]. One of the primary methods for controlling biting animals is through legislation and regulations. Local and national governments enact laws that govern pet ownership, such as licensing requirements, leash laws, and vaccination mandates. These regulations help ensure that pet owners maintain control over their animals and take responsibility for their behavior. Additionally, regulations may target specific breeds or types of animals known to pose a higher risk of biting, such as certain dog breeds with aggressive tendencies. Another important measure for controlling biting animals is public education and awareness campaigns. These initiatives aim to educate both pet owners and the general public about the importance of responsible pet ownership, including proper training, socialization, and containment of animals. Education efforts also focus on teaching children how to interact safely with animals and recognize warning signs of aggression. Animal control agencies play a crucial role in managing biting animals by responding to reports of aggressive or dangerous animals, enforcing regulations, and providing services such as pet licensing and vaccination clinics. Additionally, habitat modification and wildlife management strategies may be employed to reduce interactions between humans and wildlife species known to pose a risk of biting, such as rabies vectors. These measures may include habitat restoration, exclusion fencing, and targeted trapping or removal of problem animals. Overall, a multifaceted approach combining legislation, education, enforcement, and population management is necessary for effectively controlling biting animals and reducing the incidence of animal bites in communities [7]. By addressing both the human and animal factors contributing to biting incidents, these measures help create safer environments for both people and animals.

## 3. Dog Bites

Dog bite is by far the most frequent animal bite (about 80–90% of cases, according to Italian, American, and Australian data), followed by cat bite (about 20%) and bites from wild animals, rodents, and other pets (about 1–2%) [6,18,19,20,21]. Approximately 42% of the victims of dog bites are children and adolescents under the age of 14, particularly those in the 5–9 years-old age group, and the male/female ratio is 2:1 [22,23]. A study conducted by the National Center for Health Statistics on 84 dog-bite cases noted that pit bull species caused the most deaths (28.6%), followed by Rottweilers (19%) and German shepherds (11.9%) [1]. These data should be interpreted with consideration to the size of the animal and the resulting injuries inflicted by their bite, as small dogs will cause less extensive injuries. Even if there is a growing awareness that breed represents an unreliable indicator of risk, large dogs can of course potentially cause more severe injuries, and injuries with greater aesthetic implications, than small dogs. It should be highlighted that a child’s impulsive behavior can make the dog aggressive, causing alarm and intimidation in the animal [24]. Other factors involved are low sociality, rehearsal of prey, invasion of the dog’s territory, the child being seen as a threat, and jealousy [25]. 

Preliminary medical history represents an important part of the proper management of an animal bite. It is important to collect information about the patient’s past medical history (e.g., comorbidities such as diabetes or immunosuppression, which increase the risk of wound infection), circumstances of the incident, time passed since the bite happened, and identification of the responsible animal, as well as whether it is a pet or a stray animal [26,27]. Also, tetanus vaccination status should be investigated in order to understand whether the patient has completed the vaccination cycle and the time passed since last booster dose [28].

Analyzing the main locations of bites, in children under 5 years of age, 65% of bites are localized in the head and neck region, due to their short stature and the proximity to the dog’s mouth, but also due to the inability of the child to react defensively (the main regions bitten are the lips, nose, and cheeks) [3,29,30,31,32]. As the child grows up and increases his defensive capabilities, there is a prevalence of bites located on the limbs (in 45% of the cases there is an upper limb involvement, while in 26% of the cases there is a lower limb involvement) [33]. Most injuries do not have severe impairment; in fact about 50% of dog bites exclusively have skin involvement without injury to the underlying muscle tissues, but still require attention for cosmetic reasons [34]. 

Any bite with head, face, or neck involvement has a high risk of mortality, as these bites can result in concomitant brain trauma and hemorrhage [6,25,35]. Close attention should be paid to injuries with involvement of the cranial theca, because they can cause brain abscesses [36]. Considering that children’s skulls are thinner and less robust than those of adults, this population has a greater risk of fractures [37]. The leading cause of death from animal bites is represented by neck injury, with risk of laryngeal injury or damage to the vessels with greater hemorrhagic risk [38,39]. When the patient’s chest has been involved in a dog bite, it must be considered for rib fractures, damage to the pneumothorax, or injury to internal organs [26]. In a case series of children with severe dog-bite injuries, 8 out of 35 had a higher mortality risk when aspects like deep bites, skull fractures, pneumothorax injuries, spinal cord injuries, and tracheal and esophageal injuries were present [40]. 

Avulsion and laceration are the wounds most associated with dog bites and determined relative to anatomical features (i.e., the masseter pteroid group of the dog determines a typically deep lesion with injury to neighboring structures) [41]. The strength of the bite, depending on size and breed of the animal, causes damage to deep structures (tendons, vessels, nerves, and bones) [6,35]. Dobermanns, German shepherds and Rottweilers can apply a pressure of about 200 kg/cm^2^ [33], and with this type of injury, there should be a consultation with the plastic surgeon [13].

A deep injury is not always obvious on the first inspection, so a careful observation of the wound, considering all its characteristics, is very important, together with performing active and passive handling of the limb throughout its range of motion. Injuries with greater compressive power result in a greater risk of to damage to muscles, vessels, and nerves [33]. Teeth can inoculate germs deep inside, causing tenosynovitis and closed infections in the joints [33]. Symptoms to watch out for include pain on movement, paresthesia, hypoesthesia, typical symptoms of necrosis, and infection [26,27]. An evaluation using imaging (i.e., radiography or computed tomography) and a specialist orthopedic examination are indicated to best assess any associated injuries [28,35].

In dog bites the infectious risk is mainly related to saprophytic pathogens of the oral cavity of the animal, but also to saprophytes of the skin; the most commonly isolated strains are *Streptococcus viridans*, *Streptococcus pyogenes*, *Staphylococcus intermedius*, *Moraxella* spp., *Neisseria* spp., *Fusobaterium* spp., *Bacteroides fragilis*, *Porphyromonas* spp., *Provotella* spp., and *Pasteurella multocida* [42]. Rarely, *Capnocytophaga canimoruss* has also been isolated in dog bites; this infection, with a rapid onset within 24–48 h, presents with edema and pain on the site of inoculation, followed by maculo-papular rash, abscesses, cellulitis, and complications such as osteomyelitis, meningitis, pneumonia, and renal failure with sepsis [42]. Risk factors leading to increased mortality, with levels up to 25%, include immunosuppression and asplenia [24,43,44]. 

Dogs are the primary source of rabies transmission to humans globally [17]. Developing nations in Africa and Asia, along with parts of Latin America, continue to face significant challenges in eradicating rabies, leading to ongoing transmission cycles. The virus spreads through the saliva of infected animals, particularly through bites, scratches, or licks on broken skin. Without prompt medical intervention, rabies is almost always fatal once symptoms appear. Prompt administration of PEP, including wound care, rabies vaccine, and RIGs, can effectively prevent the onset of the disease. Therefore, immediate medical attention following any dog bite is crucial to mitigate the risk of rabies transmission, especially in high-risk countries [17].

Dog bites have a significantly lower infectious risk than cat bites (2–4% risk vs. 30–50% risk in cat bites) [33,44,45,46,47]. However, these data have been obtained from studies that are often limited and restricted in standardization. In fact, many factors can influence the infectious risk; primarily, it is determined by the type of wound, location, amount of bacteria, and presence of foreign bodies in the wound, but it is also affected by delays in the treatment, the type of dressing, and intrinsic factors that depend on the patient himself [6,36,44,48]. The hands represent a higher-risk localization, due to the presence of small compartments and lack of soft tissue compared with other body areas [49,50]. In fact, about 36% of finger lesions become infected [51]. Symptomatology of local infection must be carefully observed: erythema, swelling, fever (not always present in immunocompromised individuals), purulent exudate, and pain [26,52,53]. A study showed that treating a dog-bitten patient within 6 h after the bite leads to an infectious risk of 8% (8/95) vs. 59% (22/37) of lesions treated beyond 6 h [34]. A poorly medicated wound has a 62% infectious risk versus only 2% for a wound treated with proper debridement and irrigation [51]. 

According to recent guidelines, in the absence of signs of infection it is recommended to start a 3 to 5 day antibiotic therapy, while in cases of infection, a duration of 5 days is recommended [9]. Therapy will need to be prolonged if deeper structures are involved or if lesions are extended. Recent guidelines promote the use of antibiotics in wounds with high infectious risk, indicating the performance of a culture examination in cases in which the wound shows signs of infection or abscesses [6,42]. As described before, guidelines suggest the use of a wide-spectrum antibiotic like amoxicillin-clavulanic acid as a first-line therapy; alternatively, cefuroxime or trimethoprim-sulfamethoxazole can be used. For patients with a penicillin allergy, it is recommended to use doxycycline or a combination of clindamycin and a fluoroquinolone (clindamycin alone is not used as it does not cover for *P. multocida* infection) [6,18,36]. In addition, a tetanus vaccine is recommended if the patient was vaccinated more than 5 years ago [6]. In cases of severe and penetrating wounds, or in cases of bone and tendon exposure, hospitalization is required and the first-line treatment is represented by ampicillin-sulbactam 100–200 mg/kg/day, IV, four times daily [9]. It is important also to consider rabies PEP, based on the animal’s immunization status and the geographic areas [17].

## 4. Cat Bites

Cats are responsible for about 20–30% of all animal bites, and the mainly affected group of patients comprises adult women [4,6]. Although children are more often the victims of dog bites, cat bites also affect the pediatric population [1]. In most cases, just as with dogs, these are cats known to the patient [54]. Most cat-bite injuries are located on the upper limbs; in fewer cases, they are present on the head and neck, and, in even fewer instances, on the trunk and lower limbs [54]. In younger children, wounds are found mostly on the face and neck; as they grow, however, these bites are located more frequently in the upper limbs, and especially in the hands [6].

Cat-bite wounds seem apparently less serious than those of other types of animals, due to the smaller stretch of skin involved. However, they have one of the highest infectious rates because these types of wounds are smaller and deeper, which makes them more difficult to cleanse [4,25,55]. In their study, Jaindl et al. showed that the primary infectious rate for dog bites was 8.5%, compared to 30.6% for cats, while secondary infectious rate was almost the same (6.3% for the former vs. 6.5% for the latter) [56]. As with all wounds from animals, prompt treatment is important to prevent secondary infections [6]. The wound from a cat bite is always left open, unlike those from other animals, which remain a debated topic [4]. Even antibiotic prophylaxis, although it is still controversial for many types of animal bites, in the case of cats is always indicated because a cat bite is always considered to be at high risk of infection [25,55]. The empirical first-line treatment for children with cat bites is amoxicillin-clavulanic acid [9,25]. In children with allergic reaction to penicillins, an alternative antimicrobic regimen is represented by clindamycin plus trimethoprim-sulfamethoxazole or an extended-spectrum oral cephalosporin [4,6]. When parenteral therapy is indicated, the main choice is intravenous ampicillin-sulbactam [1].

The most frequent pathogen isolated in injuries from cat bites is *P. multocida. Pasteurella* spp. is represented by aerobes or facultative anaerobes, Gram-negative coccobacilli, which are part of the normal oral flora of 70% of cats [1,54]. The infection caused by *P. multocida* is characterized by intense pain associated with erythema and swelling due to the powerful inflammatory response of the body, which appears between 12 and 18 h after the injury. The infection usually remains localized causing cellulitis and purulent secretion. However, in some cases it can lead to abscess formation, septic arthritis, osteomyelitis, sepsis, meningitis, endocarditis, and pneumonia [1,54]. Generally, treatment with beta-lactam antibiotics is effective, although some penicillin-resistant *Pasteurella* spp. have been reported; other therapeutic options include the usage of second- and third-generation cephalosporins [8]. 

Other systemic infections transmitted by cat bites are cat-scratch disease (caused by *Bartonella henselae*), tularemia (caused by *Francisella tularensis*), sporotrichosis (caused by *Sporothrix* spp.), and rabies (caused by rabies virus) [1]. The microorganism responsible for cat-scratch disease (CSD), *B. henselae*, can be found in cat fleas that contaminate saliva, enabling the disease to be transmitted to humans [57]. CSD is more common in children than in adults and it usually starts with a blister at the injection site [58]. Subsequently, after about 1–2 weeks, ipsilateral lymphadenopathy appears (upper limbs, neck, jaw, groin, pre- and retro-auricular, clavicular, and thoracic), and it can persist for several months [58]. Sometimes there may be myalgias, arthralgias, arthritis and visceral involvement, persistent fever of unknown origin, and, more rarely, meningoencephalitis, endocarditis, and ocular involvement [59,60,61]. According to the IDSA guidelines, azithromycin is the recommended antibiotic for cat-scratch disease, with a pediatric once-daily dosage of 10 mg/kg/day for 3 days [9]. In patients weighting more than 45 kg, the dosage is 500 mg once daily for 3 days [9]. *F. tularensis* is a Gram-negative, aerobic, intracellular bacterium, and it is the etiologic agent of tularemia [62]. Tularemia is endemic in various regions worldwide, including North America, Europe, Asia, and parts of Africa [62]. In North America, it is particularly prevalent in the central and western United States, with high incidences reported in states such as Arkansas, Missouri, Oklahoma, and South Dakota. In Europe, endemic areas include Scandinavia, Eastern Europe, and parts of Russia. In Asia, tularemia is found in countries such as Japan, China, and Turkey. Additionally, cases of tularemia have been reported in parts of Africa, including North Africa and sub-Saharan regions. It is often characterized by fever, asthenia, and headache [62]. The IDSA guidelines recommend the usage of streptomycin at the dosage of 15 mg/kg twice daily by intramuscular route, or gentamycin IV at the dosage of 5 mg/kg once daily, for severe cases of tularemia. In mild cases, tetracycline and doxycycline are recommended [9]. The duration of treatment for tularemia ranges from 10 to 14 days [9,62].

Also in cat bites, depending on the geographical area, the risk of rabies virus transmission must be considered. First, the vaccination status of the animal must be assessed; if this is absent or unknown, the same protocol as for other animal bites is followed [17]. Preventive vaccination of cats is generally optional, while it is mandatory in municipalities at risk for sylvatic rabies [17].

Finally, it is mandatory to assess the patient’s immunization status against tetanus and, if necessary, proceed with the administration of the vaccine and specific immunoglobulins depending on the number of doses received by the patient and the time elapsed since those doses [6]. 

## 5. Snake Bites

Currently, available epidemiological data on snakebite are fragmentary and inaccurate. Many snakebite victims do not seek medical attention, turning instead to traditional treatment methods; the available data show an incidence of snakebite of 4.5–5.4 million people per year; of these, 1.8–2.7 million people develop clinical problems, and 81,000–138,000 people die from bite complications [63]. In tropical countries, snakebites are included among the occupational diseases affecting agricultural workers [64]. Data from the Southern Asia and Southeast Asian regions show the highest incidence and mortality [64]. In the USA, snakebites represent a minor problem, with about 9000 reported cases [65] and five deaths per year [66]. In Europe, snakebites are infrequent, with 0.22 to 1.43 cases per 100,000 population/year [67]. With regard to the pediatric population, snakebites much more frequently affect children living in low-income settings [68]. Most snakebites reported in the U.S. are attributed to pit vipers of the *Crotalidae* family [65,69]. The most common venomous snake in Italy is *Vipera aspis*, which is responsible for the most venomous bites, while in the European region the most frequent species is *Vipera berus* [67]. 

Snake venoms are the most complex of the natural venoms. The toxins in snake venom most involved in human poisoning affect the nervous, cardiovascular, and hemostatic systems, and can go so far as to cause tissue necrosis [64]. Snakebites in children require special attention because of a smaller dilution volume, resulting in a higher concentration of venom than found in adults [65]. The increased ratio of venom to body weight may result in more rapid and severe neurotoxicity, coagulopathy, and local tissue damage [70]. The severity of the wound usually depends on the size of the snake, the site of the bite, the size of the patient, the potential of the venom to cause toxic effects, and the depth of the bite, especially when the fascia have been penetrated [65]. Local swelling of the bitten region is usually more extensive and severe in the pediatric population, although it tends to heal faster than in adults, with complete healing in about 1 month [71]. However, there is a risk of developing a compartment syndrome based on the site of the bite, the volume and type of venom injected, and the local reaction to the event [68]. About a quarter of snakebites occur without inoculation of the venom, for the purpose of scaring and driving away the predator; those configure as “dry bites” [67]. The clinical presentation following snakebite is varied, e.g., poisoning by European vipers causes cutaneous symptomatology (i.e., edema, ecchymosis, and, more rarely, necrosis) that may be associated with systemic symptomatology such as gastrointestinal symptoms (i.e., nausea, vomiting, abdominalgia, and diarrhea), dyspnea, hematological changes (i.e., anemia, leukocytosis, and thrombocythemia) and hypotension [72]. In one study comparing snakebites in adults and children, the pediatric population was found to have a significantly increased risk of developing edema, ecchymosis, sweating, and dyspnea secondary to pulmonary edema [73]. Other signs and symptoms of poisoning after snakebite in children include fever, rash, diarrhea, vomiting, pain, and swelling of the affected limb that may be so significant as to cause a compartment syndrome necessitating fasciotomy. Anaphylactoid reactions, hypotension, and bronchospasm may also occur (these symptoms could also represent an acute or delayed reaction after administration of antivenom serum) [67,74]. In the Mediterranean region, neurotoxic manifestations with cranial nerve involvement (ptosis and blurred vision) may also occur after snakebite [67].

The priority during prehospital care of a snake-bitten person is to quickly transport the victim to the nearest hospital; use of traditional therapeutic methods delays presentation to medical attention, alters the original clinical appearance, and may result in bleeding, infection, gangrene, and other complications [64]. As to the pediatric population, the child should be kept calm and in a comfortable position, because a hyperdynamic state could accelerate the spread of venom. If possible, it an immobilization of the affected limb should be performed in a functional position (reducing lymphatic absorption of the poison), preferably below the level of the heart. The patient should be brought to medical attention as soon as possible, ensuring patency of airways and sufficient breathing support, and preventing aspiration of vomit or other fluids [75]. For bites of some snake species, the application of pressure immobilization bandages is useful, while for others, it is less so; moreover, if applied with mild pressure, these could increase local tissue damage. Given these considerations, the application of pressure bandages is not recommended by the WHO for most snakebites worldwide, although, nevertheless, the immobilization of the bitten limb is an essential aspect for all snakebites [68].

As for the diagnostic aspect, only a small number of high-income countries, such as Australia, have enzyme immunoassays for the identification of a specific snake venom [68]. These tests are particularly useful where there is a wide distribution of snake species across the territory, the clinical manifestations caused by the different species are similar, and a specific antivenom serum is required. At the hospital, an ABCDE approach to the patient should be undertaken initially, after which it is useful to collect anamnestic information such as the time of the event and a description of the snake (a photo of the snake, if available, is most helpful); the first-aid measures performed should then be checked, along with the patient’s general medical history and food and drug allergies [68,76]. Ensuring adequate analgesia is also very important, although poisoning from some snakes causes little or no pain; ketamine is commonly used for analgesic purposes in low- and middle-income countries. In cases of a suspected snakebite with hemotoxic characteristics, it is useful to perform a 20 min whole blood clotting bedside test; if the blood clots, a hemotoxin poisoning is unlikely. Antibiotic use remains a controversial issue, but broad-spectrum antibiotic use may be recommended, especially in tropical countries, where the incidence of bacterial infections on snakebite wounds is higher. Fasciotomy is often indicated for bites with cytotoxic effects, but is rarely necessary. Clinical evaluation of peripheral perfusion includes assessment of capillary refill time and palpation of peripheral wrists and compartment tightness. If required, fasciotomy should be performed in each relevant compartment of the affected limb; this should be performed under conditions where repletion of coagulation factors, if indicated, should be possible in a patient with coagulopathy [68,76].

The only specific antidote for the toxins contained in a snake venom is hyperimmune immunoglobulins from an animal (usually equine or ovine) immunized against the specific venom [64,67]. Snake venom immunoglobulins are included in the WHO List of Essential Medicines, meaning that they should be available in all settings where venomous snakes are present; despite this, the availability and accessibility of antivenom serums in many parts of Sub-Saharan Africa and Asia is extremely limited. As a general criterion, antivenom serum should be administered, if available, in all patients with systemic manifestations of poisoning or with severe locoregional effects of poisoning, particularly swelling, edema, and skin lesions extending more than two large joints from the bite site [68]. The dose of antivenom serum used in the pediatric population is the same as in adults, since the amount of venom injected does not depend on the size of the victim; however, the volume of saline solution in which the antivenom serum is diluted may be lessened, in order to avoid a fluid overload of the patient. Antivenom serum, especially when administered intravenously, can cause early reactions, from itching and urticaria to potentially fatal anaphylaxis. The overall risk of an antivenom serum-administration-related early reaction depends on the dose, route, and rate of administration, and ranges from about 3% to 80%, but only 5–10% of reactions are associated with severe symptomatology such as bronchospasm, angioedema, and hypotension [64]. Most reactions to antivenom serum administration can be controlled, after early detection, with intramuscular adrenaline [77]. Patients given antivenom serum should be observed closely for at least 2 h, and at the first signs of anaphylaxis, epinephrine should be promptly administered [64].

As for hospital management of viper bite, this consists in supportive care and administration of immunotherapy according to the degree of severity (grading severity score—GSS) developed by Audebert et al. [78] and later modified by Boels et al. [79]; adapting the latter classification system for pediatric ages, Marano et al. developed a pediatric-specific severity rating system (pGSS) [67] (Table 3).

When the pGSS is major or equal to 2, medically supervised administration of antivenom serum is recommended as soon as possible to prevent progression of the disease. Viperfav™ antivenom serum (Sanofi-Pasteur MSD, Lyon, France), obtained from the venom of V. ammodytes, V. berus, and V. aspis, appears to be the best choice for the treatment of viper bites in Italy and other regions where these species are widespread [67]. In the presence of severe pain, since poisoning cannot be ruled out, the patient should be monitored for at least 24 h (in order to assess the progression of symptoms). On the other hand, in the presence of moderate pain or in the absence of pain, given the absence of other signs of poisoning, a dry bite should be advocated (one in which there was no injection of venom or a minimal injection of venom), for which a 6 h observation in the emergency department is recommended. Administration of antibiotics is not mandatory for viper bites and should be decided on the basis of the patient’s history, clinical objectivity, and laboratory evidence. Corticosteroid administration is not indicated in these patients and does not reduce the duration of hospitalization, but may predispose the patient to secondary infections. Usually, in the adult population, a booster dose of anti-tetanus vaccination is administered, whereas in the pediatric population, in case of insecurity, it is necessary to check the vaccination schedule and perform an anti-tetanus antibody assay before administering a booster vaccination [67].

## 6. Rat Bites

Rat bites are only a small minority percentage of all animal bites; according to a report from 1999, there were about in 40,000 rat bites worldwide [80,81]. Rat scratches or rat bites, which are not always easily detectable at first inspection (due to the size of the animal), as well as eating food containing rat feces, can lead to some serious infections [82]. 

Rat bites can cause the homonymous fever, which is caused by two pathogens, *Streptonacillus moniliformis* and *Spirillum minus*; it is also known as “sodoku disease”. The former is more prevalent in North America, and the latter is seen principally in Asia. More than 200 cases of rat-bite fever have been reported in the United States, a number that is, however, underestimated [83]. Historically, poor people and children (the pediatric population represents about 50% of all cases) were the most affected by this disease inside the population [84]. In addition to rat bites, *S. moniliformis* can be transmitted by handling of the animal or by exposure to its secretions and saliva [83]. *S. moniliformis* infection-associated clinical presentation is characterized by non-specific symptoms, and this often does not help in early diagnosis: fever ranging from 38 °C to 41 °C lasting three to five days, rigors, frequent headaches, nausea, vomiting, myalgia, and pharyngodynia [83]. In some cases, regional lymphadenopathy is present, depending on the location of the bite; the persistence of lymphadenopathy leads to suspicion of *S. moniliformis* infection [85]. In about 50% of cases polyarthritis can occur, with the involvement of small and large joints (often the jaw and ankles) and with the pain, erythema and swelling typical of arthritis [85,86,87,88]. In some cases, polyarthralgia may persist for several years. Biopsies on skin lesions demonstrated leukocytic vasculitis [89]. Approximately 75% of patients develop maculopapular rash, petechiae, or purpura, especially in the extremities, with scaling found in 20% of cases [90]. Other features are represented by cardiac involvement with pericarditis, myocarditis, and endocarditis; systemic vasculitis; polyarthritis nodosa; meningitis; hepatitis; nephritis; pneumonia; and abscesses [91,92,93,94,95,96,97]. Untreated patients have a mortality rate of about 10%, which is mainly related to cardiac involvement, pneumonia nodosa, and septicemia. Initiating targeted antibiotic therapy allows rapid resolution of symptoms, although rash and polyarthralgia may be slow to resolve in some patients [81,91,98]. The diagnosis of *S. moniliformis* infection is based on immunochromatography, even if polymerase chain reaction (PCR) is another useful methodology [99,100,101,102]. The treatment of a rat bite consists of cleansing and the accurate disinfection of the wound. Moreover, if rat-bite fever is suspected, it is mandatory to start intravenous penicillin G or ceftriaxone for 7 days, then switch to an oral antibiotic therapy such as amoxicillin or ampicillin (in case of allergy these can be replaced by doxycycline) [84]. Alternatively, it is possible to use cephalosporins, clindamycin, carbapenems, vancomycin, and tetracyclines (which have been seen to be equally effective in antibiograms) [103,104]. Subjects with *S. moniliformis* endocarditis require a combination therapy comprising streptomycin and gentamycin for a total period of 6 weeks [105].

Additionally, rat bites can transmit diseases like leptospirosis, hantavirus, and tetanus [106,107,108,109,110]. Prompt medical attention and proper wound care are essential to prevent complications associated with these infections. Leptospirosis, a bacterial zoonotic infection caused by *Leptospira interrogans*, presents unique challenges when it affects children. While relatively uncommon in pediatric populations, compared to adult populations, leptospirosis can have severe consequences if left untreated [111,112,113,114,115,116]. Children are typically exposed to *Leptospira* bacteria through contact with contaminated water, soil, or animals, particularly rodents such as rats. The clinical presentation of leptospirosis in children can vary widely, ranging from mild flu-like symptoms to severe manifestations such as jaundice, renal failure, and pulmonary hemorrhage. Fever, headache, muscle pain, and abdominal discomfort are common early symptoms. Diagnosis in children can be challenging due to nonspecific symptoms and a lack of awareness among healthcare providers [117,118,119]. Recognition and treatment are crucial to prevent complications. Delayed diagnosis or inadequate treatment can lead to severe outcomes, including multi-organ failure and death. Therefore, a high index of suspicion is necessary, especially in endemic areas or following potential exposure to contaminated environments. Management of leptospirosis in children involves supportive care and antimicrobial therapy. Antibiotics such as amoxicillin and doxycycline are effective in treating leptospirosis if initiated early in the course of the disease [120,121]. In severe cases, hospitalization may be necessary for close monitoring and supportive interventions, including fluid resuscitation and renal replacement therapy. Prevention strategies focus on reducing exposure to contaminated environments and promoting hygiene practices [122]. Educating children and their caregivers about the risks associated with contact with rodents and contaminated water sources is essential. Measures such as wearing protective clothing, avoiding stagnant water, and maintaining rodent control in living environments can help mitigate the risk of leptospirosis. Research into pediatric-specific vaccines and improved diagnostic tools is ongoing to enhance the prevention and management of leptospirosis in children.

## 7. Bat Bites

The increasing number of interactions between humans and bats, mostly involving bat hunters and farmers (the individuals who most frequently consume bat meat), have resulted in an increased risk of infectious diseases spilling-over to humankind [123]. Bats are carriers of numerous human health threats and harbor more viruses per species than other mammals [124]. More than 60 viruses have been identified in bat tissues, including some highly pathogenic ones such as Lyssavirus, Nipahvirus, and Hendravirus [123]. In Australian bats, Australian Bat Lyssavirus (ABLV) is considered endemic [125]. Bats are the natural hosts for henipa encephalitic diseases, with human-to-human transmission and mortality rates of almost 75% [123]. Devastating spillover events of lyssaviruses and filoviruses of bat origin have been reported in the past in parts of Africa [123], including the Ebola virus outbreak in West Africa in 2014 [126]. In addition to the risk of lyssavirus transmission, several other viruses have been identified in European bat species; filoviruses represent one of the most feared families of viruses for human health, as these viruses can cause hemorrhagic fever in humans and nonhuman primates. Bats in the European region have also been associated with coronavirus infection [127,128]. Among the diseases potentially related to bat bites, rabies continues to be a public health problem worldwide, with about 75,000 human deaths per year, mostly in Africa and Asia and mostly in pediatric patients [129].

While in developing countries most of the documented case history of rabies in humans is canine in origin, in the United States and Canada during the decade 2000–2010 human rabies was mostly attributed to bat transmission [130]. In regions of North America and Western Europe, 5–10% of PEP for rabies in humans is performed following contact with bats. All bats fly, and many species are able to locate their prey and avoid obstacles using their sonar. However, when a bat is neurologically altered (as in the case of a rabies virus infection), a bat may not be able to use its sonar effectively for aerial navigation and, in a paretic state, have an increased risk of ataxia, landing, and traveling on top of a pet or human (all of which a healthy bat would avoid) [130]. While rabies vaccination is possible for dogs, controlling rabies in bats presents a difficult challenge [129]. As to the clinical presentation of rabies, while in the dog-transmitted form it is mainly characterized by encephalopathy, hydrophobia, and aerophobia, in the bat-transmitted form it most frequently presents with tremor, myoclonus, and deficits in cranial nerve, motor, or sensory function on neurological objective examination, as well as local motor or sensory disturbances [131]. The clinical course following the onset symptoms is generally rapid, and there is steady progression to death, which occurs in few days [125].

After an exposure to a bat, a decision should be made whether or not to initiate PEP for rabies, considering the type of exposure, the species of animal involved, and the local epidemiological situation [129,130]. PEP should be performed in case of established contact with the animal, and it should also be performed in cases where the person, especially if a young child or an adult person who was sleeping, is unable to confirm a contact with a bat that resulted in a bite (although there is controversy about this recommendation due to the low risk of rabies transmission related to these cases) [129]. PEP for rabies (i.e., wound cleaning, active immunization with rabies vaccine, and RIGs) is recommended after a bite or scratch caused by a bat, as well as in cases where either a mucosa or skin lesion with continuous solution has come in contact with saliva or neuronal tissues belonging to a bat [125,129]. 

In wound management of bat bites, infections such as tetanus should also be considered [125].

## 8. Monkey Bites

Monkey bites account for less than 1% of all animal bites [4]. Monkeys are the non-human primates (NHP) most frequently kept in captivity, and after two years of age they can become increasingly aggressive toward humans, with a threefold increased risk for infants [132]. Macaques (genus Macaca) are the monkeys that most frequently interface with humans [133]. In some zones of South and Southeast Asia, Hindu and Buddhist forest areas and temples have been refuges for macaque monkeys, so a contact with these animals is especially possible in those habitats, while in other places, such as Gibraltar, interactions between monkeys and humans are still possible even without temples or religious associations (because there are national parks or reserves) [133].

After a monkey bite, the wound must be washed thoroughly with povidone iodine or chlorhexidine, and it should be scrubbed for at least about 15–20 min [134]. Generally, monkey bite wounds are puncture-based and deep, so they are left to heal by secondary intention, and antibiotic prophylaxis with amoxicillin is always recommended, because of the high risk of infection. It is necessary, as in other cases, to determine the patient’s tetanus vaccination status [135]. Also, rabies virus can be transmitted by monkeys, and it is necessary to perform PEP in patients who are not immunized [135]. Macaque monkeys living in areas from North Africa to East Asia (“Old World” monkeys) are asymptomatic carriers of the Herpes B virus (*Cercopithecine herpesvirus* 1), which is a zoonotic agent that can be fatal in humans [134]. If not properly treated, this virus can lead to encephalomyelitis, with a mortality rate of about 70–80% [134,136,137]. In humans this virus, after an incubation period that can last from 2 days to 5 weeks, can present with flu-like symptoms, vesicular lesions, lymphadenopathy, and peripheral and central nervous system involvement; the final stage is encephalopathy, which can also be fatal [138,139]. While viral cultures are being performed, antiviral prophylaxis (oral valacyclovir for 14 days) is indicated if the animal is immunocompromised, if the wound is not cleaned and irrigated within 5 min after the bites, or if the wounds are particularly deep. If the patient starts to show signs of Herpes B virus infection, treatment consists of intravenous administration of acyclovir or ganciclovir [140]. Tregle et al. described the clinical case of a two-year-old child who presented to the emergency department for a bite on the head from a neighbor’s macaque monkey [132]. After cleaning the wound as previously described and administering rabies and tetanus immunoglobulins, intravenous acyclovir and intravenous antibiotic treatment with clindamycin were administered. Prior to administration, viral cultures were performed. The child, except for a febrile spike, was always well, and cultures over time became negative [132].

## 9. Conclusions

The management of common animal bites in pediatrics presents a multifaceted challenge requiring a comprehensive understanding of the epidemiology, clinical presentation, and treatment modalities associated with each specific species. Throughout this narrative review, we have explored various management strategies employed in the care of children presenting with bites from dogs, cats, rodents, and other animals. First and foremost, effective wound management is paramount in reducing the risk of infection and promoting optimal healing outcomes. This includes a thorough irrigation and debridement of the wound, assessment for tissue damage, and consideration of primary closure versus secondary intention healing, based on the nature and location of the injury. Additionally, tetanus vaccination status should be assessed and updated as necessary, and prophylactic antibiotics may be indicated in certain cases to prevent secondary infections. Furthermore, the role of rabies prophylaxis cannot be overstated, particularly in regions where rabies is endemic or following bites from high-risk animals such as bats or wild carnivores. Prompt administration of rabies PEP, including a rabies vaccine and RIGs, is essential in preventing the development of this deadly viral infection. Beyond wound care and rabies prophylaxis, considerations for the prevention of other potential complications must also be addressed. For example, the risk of CSD following cat bites or scratches underscores the importance of educating caregivers about the necessity of prompt medical evaluation and potential antibiotic therapy in high-risk cases. Similarly, the risk of transmission of other zoonotic pathogens such as leptospirosis or tularemia should be considered based on the circumstances of the bite and the local epidemiology. In addition to medical management, psychosocial support for both the child and their caregivers is integral to the overall care continuum. Animal bites can evoke fear and anxiety, and addressing these emotional aspects can facilitate the healing process and reduce the risk of long-term psychological sequelae. Psychosocial support for the child should focus on creating a safe space for them to express their emotions, validating their experiences, and teaching coping strategies to manage anxiety and fear. Play therapy, art therapy, or cognitive–behavioral techniques tailored to their age can be effective in addressing their psychological needs. Caregivers, on the other hand, may experience guilt, worry, and a sense of helplessness after their child is bitten by an animal. They may blame themselves for not being able to prevent the incident or worry about their child’s physical and emotional well-being. Providing caregivers with education about animal behavior, first aid, and strategies to prevent future incidents can help alleviate their anxiety. Additionally, offering them a supportive environment in which to share their feelings and concerns can validate their experiences and strengthen their ability to support their child effectively.

While the management of animal bites in pediatric populations is crucial, emphasizing prevention is paramount in reducing their incidence and associated burdens. Teaching children how to interpret animals’ body language, such as signs of fear or aggression, empowers them to recognize potentially risky situations and take appropriate action, such as seeking adult supervision or avoiding contact altogether. Additionally, promoting responsible pet ownership practices, such as proper training, socialization, and vaccination of pets, can significantly reduce the risk of bites within households. 

Overall, the latest research progress in animal-bite management in pediatrics underscores the importance of a multidisciplinary approach that addresses not only the physical aspects of wound care but also the psychological and preventive aspects to optimize outcomes and promote the well-being of pediatric patients. Looking ahead, future studies exploring the efficacy of novel treatment modalities, such as topical antimicrobial agents or advanced wound dressings, may offer new insights into optimizing wound healing and reducing the risk of complications. Antimicrobial peptides (i.e., naturally occurring molecules found in the innate immune system of many organisms, including humans with broad-spectrum antimicrobial activity) have shown promising results in preclinical studies as topical agents for the treatment of animal bites. In addition, researchers are exploring innovative drug-delivery systems to enhance the efficacy of existing treatments for animal bites in children. This includes the development of sustained-release formulations which prolong the release of antimicrobial agents at the site of the wound, ensuring continuous protection against infection while minimizing the need for frequent applications. Moreover, the evaluation of the efficacy of different types of psychosocial support for the child and the caregiver is extremely important. In addition, implementing regulations and enforcing leash laws in public spaces can minimize interactions between children and potentially aggressive animals, such as stray dogs. Furthermore, advocating for wildlife conservation and habitat preservation helps to create environments that mitigate encounters between children and wild animals, thereby reducing the likelihood of bites and wildlife-related injuries.

## Figures and Tables

**Table 1 microorganisms-12-00924-t001:** Schematic antitetanic management in Italy according to the Italian Health Ministry.

Vaccination Status	Superficial and Clean Wounds	All Other Wounds
Complete vaccination cycle, with last vaccination less than 5 years ago	No vaccination required	No vaccination required (booster only for high-risk wound). No anti-tetanus immunoglobulin required.
Complete vaccination cycle, with last vaccination between 5 and 10 years ago	One booster required (DPT)	One booster required (DPT). No anti-tetanus immunoglobulin required.
Complete vaccination cycle, with last vaccination more than 10 years ago	One booster required (DPT)	One booster required (DPT). Anti-tetanus immunoglobulin required.
Uncompleted or uncertain cycle	Start vaccination cycle	Start vaccination cycle and anti-tetanus immunoglobulin.

DPT = Diphtheria–Pertussis–Tetanus.

**Table 2 microorganisms-12-00924-t002:** Recommendations regarding rabies vaccines.

Category of Exposure	Type of Exposure	Recommendations
Category I	Touching or feeding animals, animal licks on intact skin (no exposure)	Wound should be immediately washed thoroughly with soap and water for at least 15 min. No PEP required
Category II	Nibbling of uncovered skin, minor scratches, or abrasions without bleeding (exposure)	Wound should be immediately washed thoroughly with soap and water for at least 15 min. Immediate vaccination at day 0, then additional doses at days 3 and 7 (if necessary) RIG is not indicated
Category III	Single or multiple transdermal bites or scratches, contamination of mucous membrane or broken skin with saliva from animal licks, exposures due to direct contact with bats (severe exposure)	Wound should be immediately washed thoroughly with soap and water for at least 15 min. Immediate vaccination at day 0, then additional doses at days 3 and 7 and between days 14–28 RIG administration is recommended at day 0

PEP = post-exposure prophylaxis, RIG = rabies-specific immunoglobulin.

**Table 3 microorganisms-12-00924-t003:** Pediatric-specific severity score for management of viper bite.

Pediatric Grading Severity Score (pGSS)	Characteristics	Suggested Interventions
0 → NO ENVENOMING/DRY BITE	Fang marks; No edema and no local reaction.	6 h surveillance in the emergency room.
1 → MINIMAL ENVENOMING	Local edema around the bite area; No systemic symptoms.	Clinical observation up to reduction of edema; Supportive care, including hydration and pain relief.
2 → MODERATE ENVENOMING	2a → One or both of the following: Regional edema with progression to most of the limb; Hematoma or adenopathy. 2b → Grade 2a+ moderate general symptoms (mild hypotension, vomiting, diarrhea, neurotoxic signs) and/or biologic criteria for severity: Leucocytes > 11,000/L, Neutrophils > 65%, INR > 1.15.	Clinical observation up to the evident reduction of edema; Supportive care, including hydration and pain relief; Doppler ultrasound of affected limb’s blood vessels; Administration of antivenom; Evaluate antibiotic therapy (only if clinical or laboratory signs of bacterial contaminations are evident); Administer LMWH if direct evidence of thrombophlebitis or in cases of extensive edema, dehydration, decreased mobility, prolonged decubitus, admission to PICU, anticipated hospitalization > 48 h. Do not administer in cases of overt hemorrhage or a bleeding disorder
3 → SEVERE ENVENOMING	One or both of the following: Edema spreading to the trunk; Signs of hemodynamic instability (prolonged hypotension, shock, bleeding).	Same interventions as in grade 2; Admission to PICU.

Adapted from Audebert–Boels classification [78,79] and Marano et al. [67]. LMWH, low molecular weight heparin; PICU, pediatric intensive care unit.

## Data Availability

Not applicable.

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
