# Peer review of "Management Strategies for Common Animal Bites in Pediatrics: A Narrative Review on the Latest Progress"

_microorganisms, 2024, doi:10.3390/microorganisms12050924_

Round 1

Reviewer 1 Report

Comments and Suggestions for Authors

The manuscript being reviewed is meticulously organized and provides a comprehensive account of animal bites, including valuable information on diagnosis and management. The authors have taken into consideration the history, epidemiology, wound management, and treatment modalities. However, to ensure the review is truly insightful and covers new approaches, it needs to be described in more depth and provide insight into noble diagnosis and treatment. The manuscript presents two promising areas of research, psychosocial treatment and the development of new antimicrobial agents and wound dressings, which should be the authors' focus. Delving deeper into these aspects will enable the manuscript to make a more significant contribution to the existing literature on animal bites.

Question 1. How and what kind of psychosocial support would you suggest? Is it practical and effective?

Question 2. What kind of dressing would you specify for the efficient treatment? 

Question 3. Do you have any insight on topical antimicrobial agents or new drug applications? 

Comments on the Quality of English Language

The overall quality of the English language used in this manuscript is fine, with only minor adjustments required.

Author Response

The manuscript being reviewed is meticulously organized and provides a comprehensive account of animal bites, including valuable information on diagnosis and management. The authors have taken into consideration the history, epidemiology, wound management, and treatment modalities. However, to ensure the review is truly insightful and covers new approaches, it needs to be described in more depth and provide insight into noble diagnosis and treatment. The manuscript presents two promising areas of research, psychosocial treatment and the development of new antimicrobial agents and wound dressings, which should be the authors' focus. Delving deeper into these aspects will enable the manuscript to make a more significant contribution to the existing literature on animal bites.

Re: Thank you for your evaluation. We revised the manuscript according to your suggestions.

Question 1. How and what kind of psychosocial support would you suggest? Is it practical and effective?

Re: We added some paragraphs on this issue in the Conclusions (pp. 16-17).

Question 2. What kind of dressing would you specify for the efficient treatment?

Re: We added some paragraphs on this issue (p. 2).

Question 3. Do you have any insight on topical antimicrobial agents or new drug applications?

Re: We added some paragraphs on these insigts in the Conclusions.

Reviewer 2 Report

Comments and Suggestions for Authors

This narrative review offers a thorough examination of the prevailing management approaches for common animal bites in pediatric patients, delving into the epidemiology, clinical manifestations, and treatment options linked to various animal species. It underscores the complex challenges these injuries pose in children.

However, two notable issues require attention.

Firstly, much of the content discussed in this review is considered basic knowledge among clinicians and fails to incorporate recent research advancements. Given the breadth of the topic of animal bites, encompassing everything within a single article is challenging. Consequently, the overall content appears somewhat superficial, lacking in-depth discussions, comparisons, and analyses. It is recommended that the author revisit the title and revise the manuscript to align it with the latest research progress, emphasizing evidence-based medical findings that offer reliable insights.

Secondly, while the review predominantly focuses on the medical management of animal bites, it inadequately addresses the significance of prevention. Considering the considerable morbidity and potential fatality associated with animal bites, especially in children, it is imperative to prioritize strategies for preventing these injuries. Future studies should explore the efficacy of educational interventions and community-based programs aimed at reducing the occurrence of animal bites among pediatric populations.

Author Response

This narrative review offers a thorough examination of the prevailing management approaches for common animal bites in pediatric patients, delving into the epidemiology, clinical manifestations, and treatment options linked to various animal species. It underscores the complex challenges these injuries pose in children.

Re: Thank you for your comments. We revised our manuscript accordingly.

However, two notable issues require attention.

Firstly, much of the content discussed in this review is considered basic knowledge among clinicians and fails to incorporate recent research advancements. Given the breadth of the topic of animal bites, encompassing everything within a single article is challenging. Consequently, the overall content appears somewhat superficial, lacking in-depth discussions, comparisons, and analyses. It is recommended that the author revisit the title and revise the manuscript to align it with the latest research progress, emphasizing evidence-based medical findings that offer reliable insights.

Re: We improved the manuscript with in-depth discussion in the Introduction (pp. 1-2), in the chapter on General management of animal bite wounds (pp. 2-5) and in the Conclusions (pp. 16-17). The Title has been changed according to your recommendation (p. 1).  

Secondly, while the review predominantly focuses on the medical management of animal bites, it inadequately addresses the significance of prevention. Considering the considerable morbidity and potential fatality associated with animal bites, especially in children, it is imperative to prioritize strategies for preventing these injuries. Future studies should explore the efficacy of educational interventions and community-based programs aimed at reducing the occurrence of animal bites among pediatric populations.

Re: Prevention has been largely added in the manuscript (pp. 2, 5, 16).

Reviewer 3 Report

Comments and Suggestions for Authors

Dear authors

Thanks for your presentation.

However, some comments should be considered during your revision.;

  1. Some statements in the abstract are repeated in the other parts in the manuscript (lines 11-12 are similar to line 39-40).
  2. It is not referable to add the statement in lines 11-12 (that contains percentages) in the abstract.
  3. Most of references are old. They should be updated (the last five years).
  4. References section should checked (eg. references 73, 74).
  5. The introduction section is too short.
  6. Line 51, “abstracts”.
  7. Line 102, Staphylococcus aureus should be in italic.
  8. Since the abbreviations “MRSA and PCR’ have been mentioned for one time, it should be deleted.
  9. The immunoglobulin in Table (1) should be specified either in the table or in the legend (Ig for what?).
  10. In section “General management of animal bite wounds”, the authors mentioned the different methods used for controlling the animals bites, what about the animal itself? What are the measurements used for controlling the biting animals?
  11. The site of biting (head, hands, limbs, etc.) should be taken in consideration during handling of bitted case. Bites near the brain (Rabies) should be treated immediately.
  12. The treatment strategy has been repeated in dog bites section (Lines 228-243, 264-269).
  13. The most important pathogens isolated from dogs bits and their signs in the bitted patient should be mentioned (as in cat’s bites section).
  14. Line 278, “Pasteurella” should be italic.
  15. Line 281, the abbreviation “F. tularensis” should follow the full name of the bacterium.
  16. Lines 294-300, the world-wide distribution of tularemia should be supported with references.
  17. Most of the mentioned bites are of rare incidences “occupational”, except dog and cat bites.  

Best wishes

Author Response

Dear authors,

Thanks for your presentation. However, some comments should be considered during your revision.

Re: Thank you for your suggestions. We revised the manuscript according to your recommendations.

Some statements in the abstract are repeated in the other parts in the manuscript (lines 11-12 are similar to line 39-40).

Re: Revised as suggested (p. 1).

It is not referable to add the statement in lines 11-12 (that contains percentages) in the abstract.

Re: The percentages have been deleted from the abstract (p. 1).

Most of references are old. They should be updated (the last five years).

Re: As reported in the paragraph that described our search strategy (p. 2), we included all the manuscripts published in the last 30 years.

References section should checked (eg. references 73, 74).

Re: All the references have been carefully revised (pp. 15-25).

The introduction section is too short.

Re: Revised as suggested (pp. 1-2).

Line 51, “abstracts”.

Re: Corrected (p. 2).

Line 102, Staphylococcus aureus should be in italic.

Re: Done (p. 3).

Since the abbreviations “MRSA and PCR’ have been mentioned for one time, it should be deleted.

Re: Done (p. 3).

The immunoglobulin in Table (1) should be specified either in the table or in the legend (Ig for what?).

Re: Clarified (pp. 3 and 5).

In section “General management of animal bite wounds”, the authors mentioned the different methods used for controlling the animals bites, what about the animal itself? What are the measurements used for controlling the biting animals?

Re: A paragraph on measurements for controlling the biting animals has been added (p. 5).

The site of biting (head, hands, limbs, etc.) should be taken in consideration during handling of bitted case. Bites near the brain (Rabies) should be treated immediately.

Re: Added as suggested (p. 4).

The treatment strategy has been repeated in dog bites section (Lines 228-243, 264-269).

Re: We think that it is useful to summarize the treatment strategy at the end of each chapter related to specific animals.

The most important pathogens isolated from dogs bits and their signs in the bitted patient should be mentioned (as in cat’s bites section).

Re: They are reported (p. 7).

Line 278, “Pasteurella” should be italic.

Re: Done (p. 8).

Line 281, the abbreviation “F. tularensis” should follow the full name of the bacterium.

Re: Done (p. 8).

Lines 294-300, the world-wide distribution of tularemia should be supported with references.

Re: Added (p. 9).

Most of the mentioned bites are of rare incidences “occupational”, except dog and cat bites. 

Re: It is written in the Introduction and the Conclusions that dog and cat bites are the most common, but in our experience there is an increase of bites due to other animals in pediatrics. For this reason, we decided to perform this comprehensive review.

Round 2

Reviewer 2 Report

Comments and Suggestions for Authors

Both issues raised during the last review have been resolved. I have no further suggestions.